# Optimization of Anthocyanin Production in Tobacco Cells

**DOI:** 10.3390/ijms241813711

**Published:** 2023-09-05

**Authors:** Andrea Carpi, Md Abdur Rahim, Angela Marin, Marco Armellin, Paola Brun, Giovanni Miotto, Renzo Dal Monte, Livio Trainotti

**Affiliations:** 1Active Botanicals Research (ABR), 36040 Brendola, Italy; andrea.carpi@abres.it (A.C.); a.marin@hyperionph.com (A.M.); renzo.dalmonte@abres.it (R.D.M.); 2Department of Biology, University of Padua, 35131 Padua, Italy; rahimgepb@sau.edu.bd (M.A.R.); marco.armellin.2@studenti.unipd.it (M.A.); 3Department of Genetics and Plant Breeding, Sher-e-Bangla Agricultural University, Dhaka 1207, Bangladesh; 4Department of Molecular Medicine, University of Padua, 35131 Padua, Italy; paola.brun.1@unipd.it (P.B.); giovanni.miotto@unipd.it (G.M.); 5Botanical Garden, University of Padua, 35123 Padua, Italy

**Keywords:** *Nicotiana tabacum*, *Prunus persica*, synthetic biology

## Abstract

Plant cell cultures have emerged as a promising tool for producing active molecules due to their numerous advantages over traditional agricultural methods. Flavonols, and anthocyanin pigments in particular, together with other phenolic compounds such as chlorogenic acid, are known for their beneficial health properties, mainly due to their antioxidant, antimicrobial, and anti-inflammatory activities. The synthesis of these molecules is finely regulated in plant cells and controlled at the transcriptional level by specific MYB and bHLH transcription factors that coordinate the transcription of structural biosynthetic genes. The co-expression of peach *PpMYB10.1* and *PpbHLH3* in tobacco was used to develop tobacco cell lines showing high expression of both the peach transgenes and the native flavonol structural genes. These cell lines were further selected for fast growth. High production levels of chlorogenic acid, anthocyanins (mainly cyanidin 3-rutinoside), and other phenolics were also achieved in pre-industrial scale-up trials. A single-column-based purification protocol was developed to produce a lyophile called ANT-CA, which was stable over time, showed beneficial effects on cell viability, and had antioxidant, anti-inflammatory, antibacterial, and wound-healing activities. This lyophile could be a valuable ingredient for food or cosmetic applications.

## 1. Introduction

Anthocyanins are plant-derived, water-soluble, and bioactive secondary metabolites and are liable for the coloring of different plant organs like leaves, stems, roots, flowers, fruits, and grains as red, purple, and blue colors [1,2,3]. They are the flavonoid class of polyphenols and biosynthesized through the phenylpropanoid pathway starting from the essential amino acid phenylalanine. The structural genes involved in anthocyanin biosynthesis are well characterized in several plant species. In dicot, these structural genes are coordinately regulated by three protein families, including R2R3-MYB transcription factors (TFs), bHLH TFs, and WD40 repeat proteins through the formation of a transcriptional complex (MBW), where MYB TFs play the main regulatory role in anthocyanin biosynthesis (Figure 1) [4,5,6]. Anthocyanins play an important role in growth, development, reproduction, and responses to various environmental (abiotic and biotic) stresses on plants [7]. They act as powerful antioxidants with definite biological activities, including anticancer, antimicrobial, antiaging, and antidiabetic agents, and the prevention of heart diseases [1,8,9,10]. They are also beneficial to humans, as anthocyanin pigments are used in the food processing, pharmaceutical, and cosmetics industries [1]. Consequently, the potential application of such health-promoting anthocyanin pigments is increasing in the food, cosmetics, natural colorants, and pharmaceutical industries. However, the large-scale production and extraction of anthocyanin pigments are still inadequate.

Plant cell cultures have emerged as a promising tool for producing active molecules due to their numerous advantages over traditional methods. This technology efficiently produces the desired compounds through plant cell cultures in a controlled environment and offers several advantages in producing active molecules. It provides a consistent and sustainable supply of bioactive compounds independent of seasonal variations and plant rarity. Plant cell cultures eliminate the need for extensive land usage and dramatically reduce water consumption, avoiding pesticides, fertilizers, and other agronomic interventions potentially harmful to the environment. Furthermore, this technology allows production optimization through genetic engineering and process development, resulting in higher yields and the purity of active molecules [11,12]. Plant cell cultures have found applications in diverse fields, including nutraceuticals, pharmaceuticals, and cosmetics. This technology is used in the pharmaceutical industry for the production of anticancer drugs [13]. In nutraceuticals, suspension cell culture technology has produced food supplements to support human health [14]. In cosmetics, active molecules derived from plant cell cultures are used in skincare and hair care products [15]. Previous reports have described these achievements, including examples of *Perilla frutescens*, *Vitis vinifera*, *Daucus carota*, *Ajuga pyramidalis*, *Vaccinium pahalae*, etc., for large-scale anthocyanin production [16,17,18,19,20].

The genetic engineering of key regulators MBW/structural genes enables the production of enhanced levels of anthocyanin in plant cells [11,21]. In peach, *PpMYB10.1* and *PpbHLH3* have been shown to be responsible for anthocyanin accumulation in the fruit during ripening [6], and PpMYB10.1 overexpression in tobacco cells induces enhanced anthocyanin pigmentation [6,22].

In the present study, we obtained tobacco cells expressing the peach PpMYB10.1 and PpbHLH3 TFs and optimized the large-scale production of anthocyanins in tobacco cells in a bioreactor for the industrial production of valuable polyphenols, which were tested for some of their possible beneficial activities.

## 2. Results

### 2.1. Development of Tobacco M1-B3 Over-Expressing Cell Lines

#### 2.1.1. Development of Tobacco M1-B3 Plant Clones

Tobacco plants expressing the peach PpMYB10.1 (M1 hereafter) TF were already available in the laboratory [22], as well as genetic constructs for the expression of PpbHLH3 (B3 hereafter) [6]. Several independent clones of tobacco plants carrying B3 driven by the pOp promoter, recognized by the synthetic transcription factor LhG4 [23], were obtained following standard protocols. As expected, none of them was significantly different from the WT, as the 35S:LhG4 driving construct [23] was absent (data not shown). Pollen from several of these B3 lines was used to cross-pollinate “Type II (mild phenotype)” M1 over-expressing lines, able to produce fertile ovules [22] (Figure 2A). Seeds from these crosses were plated on wet paper in petri dishes and individual accumulating pigments already in cotyledons were easily selected (Figure 2B) to be moved onto soil (Figure 2C). Soil-grown individuals were PCR-checked for both M1 and B3 in their genomes. Among the double-positive clones, even individuals from the same cross displayed different levels of pigment accumulation (Figure 2D), which also varied depending on the growing conditions (data not shown). As for clones expressing only M1 [22], flowers were the sites where pigment accumulation was more visible (Figure 2E,F).

#### 2.1.2. Development of Tobacco M1-B3 Plant Cell Lines

Leaf fragments from several M1B3 tobacco pigmented plants were used to develop cell culture lines on solid media (Figure 3A–C). Repeated selection of the darker cells on a solid medium allowed the isolation of the best clones (Figure 3C) from which liquid cultures were developed. Pilot scale-up experiments in up to 25 L bioreactors allowed us to reach optimal results after 2 weeks of culture. At this stage, both control (cells carrying only the 35S:LhG4 driving construct; Figure 3D) and M1B3 (Figure 3E) cells were only partly aggregated and had a rod shape, but only the latter accumulated vacuolar pigments.

#### 2.1.3. Molecular Characterization of Tobacco M1B3 Plant Cell Lines

Expression profiles of the peach M1 and B3 transgenes and the tobacco *NtDFR* and *NtUFGT* target genes, taken as examples of all structural genes transactivated by the MBW complex, were tested in tobacco cell lines and compared to their levels in the flowers and leaves of their greenhouse-grown parental plants. M1 levels were higher in the cell cultures than in both the flowers and leaves of the M1B3-expressing clone but undetectable in controls (a genotype carrying only the 35S:LhG4 driving construct). On the contrary, B3 expression was similar in both cell cultures and the organs of the greenhouse-grown plants. Also, B3 expression was undetectable in the control genotype (Figure 4). The expression of the two peach TFs was able to increase at least by 10-fold of the transcription of both *NtDFR* and *NtUFGT* in both cell cultures and greenhouse-grown plants. In controls, *NtDFR* and *NtUFGT* expression was barely detectable but slightly higher in flowers (Figure 4).

### 2.2. Chemical Characterization of the Polyphenol Extracts

#### 2.2.1. HPLC Characterization of the Polyphenol Extracts

In order to verify the effect of the peach TFs on the biosynthesis of anthocyanins and polyphenols, the total extracts of the freeze-dried leaves of the control and M1B3 plants were analyzed by HPLC. In the control plant leaf extract, the chromatographic profile revealed a low concentration of polyphenols (evaluated at λ = 330 nm) and extremely low levels of anthocyanins (evaluated at λ = 514 nm; Figure 5A,B). On the contrary, in the leaves of the transgenic plants, a marked increase in polyphenols (60–80 mg/g) was observed, and a peak attributable to anthocyanins appeared (about 0.1 mg/g; Figure 5C,D). In the stabilized and selected cell line, the production of polyphenols and anthocyanins was exacerbated. In particular, the polyphenol that appeared to be predominant was chlorogenic acid and the anthocyanins underwent an increase in synthesis, reaching about 25 mg/g of the freeze-dried cells (Figure 5E,F). Compared to the transgenic leaves, the anthocyanins showed an increase in concentration of about 250 times that in the cell line (normalization on freeze-dried dry weight) of the same genotype.

Because transgenic cells showed a good growth rate (doubling of biomass every 5–7 days), the culture process was optimized at the pre-industrial level in 25 L bioreactors. In order to semi-purify the anthocyanins, the cell extract at the end of the bioreactor culture period was loaded into a column containing a specific adsorption resin. The resin completely traps the anthocyanins, and these could be concentrated and freeze-dried after elution. Interestingly, the resin also efficiently trapped chlorogenic acid, which co-eluted and concentrated with anthocyanins. The lyophile obtained from this process contained about 10% anthocyanins and about 30% chlorogenic acid (Figure 5G,H). For this reason, it was named ANT-CA. Usually, the remaining fraction of the lyophile in these semi-purification processes consists of carbohydrates derived from plant cells (a more accurate characterization of this fraction will be necessary).

#### 2.2.2. Anthocyanin Characterization by HPLC–Mass Spectrometry

The lyophilized ANT-CA powder contained anthocyanins derived from cyanidin, delphinidin, and petunidin. The most abundant are reported in Table 1.

Compounds related to pelargonidin, peonidin, and malvidin seem absent in the lyophilized powder.

The lyophilized powder contains, in addition to anthocyanins, chlorogenic acid (rt 2.85 min), caffeoyl and feruloyl putrescine (2.6 and 1.8 min respectively), and some glycosylated hydroxy flavanone compounds that need to be confirmed by comparison with a known reference material.

### 2.3. Cell Viability Assay with the ANT-CA

To exclude the toxic effects of ANT-CA on cells and identify the experimental concentrations, we performed a cellular viability assay using the MTT test. ANT-CA significantly decreased THP-1 and BJ cell viability starting at 150 μg/mL (Figure 6) compared with non-treated cells. Thus, ANT-CA at the concentration of 100 μg/mL was used in subsequent experiments in THP-1 and BJ cells.

### 2.4. Incubation with ANT-CA Quenches Intracellular Reactive Oxygen Species Production

The incubation of THP-1 or BJ cells with ANT-CA for 2 or 4 h did not affect intracellular reactive oxygen species (ROS) production. Intracellular ROS levels drastically increased in THP-1 and BJ cells following incubation with H_2_O_2_. The pretreatment of cells with ANT-CA for 2 and 4 h significantly reduced ROS levels in cells, thus supporting the antioxidant activities of ANT-CA (Figure 7).

### 2.5. ANT-CA Reduces Lipopolysaccharide-Induced Inflammation in THP-1 Cells

Intracellular ROS act as transducing molecules in mammalian cells and engage inflammatory signals in differentiated THP-1 cells [17]. To further evaluate the role of ANT-CA in macrophage-like cells, pre-treated THP-1 cells were incubated with lipopolysaccharide (LPS) to simulate a pro-inflammatory insult. As reported in Figure 8, incubation with LPS significantly increased the production of IL-1β and IL6. Pretreatment with ANT-CA dampened the LPS-induced cytokine production. In particular, in THP-1 cells treated with ANT-CA, the concentration of IL-1β in the conditioned media regained values observed in non-treated cells (Figure 8).

### 2.6. ANT-CA Reports In Vitro Wound Healing Potential

Fibroblast-like cells play a central role in tissue repair. To assess the potential of ANT-CA in supporting wound closure, we wounded BJ cell monolayers and evaluated the wound healing time. Experiments were stopped at 72 h of incubation when we still observed gaps in the wounds of non-treated cells. As reported in Figure 9A, ANT-CA-treated cells migrated beyond the margins of the wound and populated the gap within 72 h; in non-treated cells, empty spaces remained inside the wound after 72 h of incubation.

Following 72 h of incubation, the number of cells crossing the depicted lines was recorded within a minimum of five randomly selected fields covering 1.254 mm^2^ from at least three independent experiments. The analyzed area was estimated using NIH Image J software (https://imagej.en.softonic.com/ (accessed on 21 July 2023)). The data are reported in Figure 9B.

### 2.7. ANT-CA Demonstrates a Low Antibacterial Activity as Compared to Antibiotics

Incubation with ANT-CA reported antibacterial activity against MRSA and *Pseudomonas aeruginosa*, as demonstrated by bacterial growth inhibition zones around the wells. The diameters of the clear zones were measured and the data are reported in Figure 10. ANT-CA demonstrated better antibacterial activity against the Gram-negative bacteria (*P. aeruginosa*) than the Gram-positive bacteria (MRSA). However, when we compared the data with the inhibition zones generated by the respective antibiotics, the antibacterial activities of ANT-CA were low (Figure 10).

## 3. Discussion

Generally, M1B3 plants were more pigmented than those previously described carrying M1 alone, which looked green despite several growth impairments [22]. Although not studied in detail, M1B3 clones also showed several growth defects, such as shorter stamens or reduced seed setting (data not shown) but, as these were already present in M1 plants [22], they are ascribable to the action of only the PpMYB10.1 TF. On the contrary, as previously shown with transient assays [6], the two peach TFs, when present together at the relatively high levels assured by the LhG4/pOp expression system [23], were very active in transactivating the tobacco structural genes of the phenylpropanoid pathway and, more importantly, not only in parts where tobacco usually produces these pigments, such as the corolla or the seed coat, but also, among others, in leaves. This leads us to think about the possibility of the biotechnological exploitation of this plant material, given that polyphenols, and anthocyanins in particular, are valuable molecules [24].

Cell lines developed from individuals that were the darker among those grown under standard greenhouse conditions varied in pigment accumulation. Moreover, several cycles of the selection of the more pigmented cells among those of the growing calluses were necessary to develop the line that was used for the industrial scale-up. These variations are in agreement with previous findings, as shown, among others, in tobacco callus cultures over-expressing the Arabidopsis *PAP1* gene [25], and likely depend on the complex network of activators and repressors regulating the flavonoid pathway [21]. This selection most likely favored the cell lines in which repressors of the pathway were less active [26], but more detailed analyses will be necessary to clarify the molecular reasons for the observed non-reversions. Nevertheless, in the pigmented lines, high expression levels of both the peach transgenes and the tobacco target genes were observed. Here, we report only the testing of *NtDFR* and *NtUFGT*, as the ability of the two peach TFs to transactivate other tobacco structural genes such as *NtPAL*, *NtCHS*, *NtCHI*, *NtF3H*, *NtLAR*, and *NtASN* has been already shown in previous works [6,22] and is in agreement with the observation of the AmDel/AmRos1 expression in tomato [8,27] and tobacco [11]. Again in agreement with other reports [8,11,27], the expression of the two TFs led not only to anthocyanin accumulation but also to increases in chlorogenic acid, accounting for about 30% of the ANT-CA lyophile composition. This molecule is very interesting for its therapeutic roles, as it has, among others, antioxidant, antibacterial, hepatoprotective, cardioprotective, and anti-inflammatory activities and, since it is a free radical scavenger, neuroprotective and anti-hypertension roles [28]. As regards the different anthocyanins detected, cyanidin 3-rutinoside was by far the most abundant, as in other cell-based expression systems [11]. Considering the ABResearch bioreactors with a volume of 1000 L, the production of about 800 g of lyophile (80 g of cyanidins and 240 g of chlorogenic acid) could be achieved for each operating unit.

The ANT-CA lyophile, confirming the beneficial effects of polyphenols on cell viability, has antioxidant, anti-inflammatory, and antibacterial activities [24,29,30] and also showed a wound healing function, thus confirming that several formulations containing anthocyanins can ameliorate the symptoms and speed up recovery after wounding [31,32].

In the food industry, anthocyanins represent a safer alternative to synthetic colorants [11]. Due to their vivid colors and biological properties, anthocyanins have been regulated in the cosmetic industry [33], where natural and sustainable ingredients are highly valued and offer an alternative to synthetic dyes. Besides their color properties, the anthocyanins in ANT-CA possess skin-friendly properties. They are known to exhibit anti-aging effects by reducing the formation of reactive oxygen species and protecting the skin by shielding UV light. The antioxidant properties of anthocyanins could help to protect the lips from oxidative damage and maintain their health and appearance. In addition, inflammatory conditions such as irritated lips could benefit from the anti-inflammatory effects of both anthocyanins and chlorogenic acid. Chlorogenic acid has photoprotective effects against UV radiation, which can cause damage to the lips, including sunburn, dryness, and the development of fine lines. In addition, chlorogenic acid has hydrating properties that can help improve the lips’ moisture content, keeping them healthy and preventing dryness and cracking. Anthocyanins and chlorogenic acid are generally considered safe for topical use, and their combination could allow the production of colored lipsticks in which the synergistic effect of the two molecules protects and maintains the health of the lips.

## 4. Materials and Methods

### 4.1. Explant Source and Surface Sterilization

Leaf explants of apparently healthy grown plants were collected. At first, the plant materials were washed under running tap water and then soaked in 70% ethanol for 1 min. The explants were then rinsed three times in sterile distilled water and inoculations of the explants were performed aseptically in a laminar flow hood.

### 4.2. Callus Cultures Establishment

Callus cultures were initiated on solid standard Murashige and Skoog medium (MS) comprising 30 g/L sucrose, 4.4 g L of MS powder (Duchefa), and 1 g/L agarose before autoclaving. The pH of the media was adjusted to 5.8. Explants were cultured in plates incubated in dark conditions at 25 °C. The morphology of the calluses such as color, rate of growth or reversion, and friability, was evaluated over time.

### 4.3. Callus Culture Selection and Liquid Growth Conditions

After establishment, calluses were grown in different media for 6 months to identify the best one in which cells grew better and in a disaggregated way, maintaining the dark violet color, an indicator of the expression of the genes of interest. The media selected was Gamborg B5 basal medium, with the addition of phytohormones such as IAA (indole acetic acid) 0.2 mg/L, NAA (naphthaleneacetic acid) 1 mg/L and Kinetin 1 mg/L, and sucrose 30 g/L and 1 g/L agarose for solid medium. The same basal composition was used for liquid cultures. The cells were grown at 25 °C in dark conditions, and suspension cultures were conducted at 110 rpm. Because of the rapid growth, cells were transferred every 7 days to a fresh medium by selecting only violet cells. Cell suspensions were then scaled up to a bioreactor of 25 L.

### 4.4. Total RNA Extraction and qRT-PCR Analyses

Total RNA was extracted from tobacco leaves, flowers, and suspension cell cultures according to the Chang protocol [34]. RNA purity and integrity were checked by UV absorption spectra agarose electrophoresis. Before cDNA synthesis, RNA was treated with DNAse I (Sigma-Aldrich). Treated RNA was retrotranscribed using the High-Capacity cDNA Reverse Transcription Kit (Applied Biosystems). Expression analyses were carried out by real-time PCR with the kit GoTaq^®^ qPCR Master Mix (Promega), using UBIQUITIN as the internal standard. Primer sequences used for the qRT-PCR analyses are listed in Table 2. The obtained CT values were analyzed by means of the “Q-gene” software (https://www.qgene.org/qgene/download.php (accessed on 22 July 2023)) [35] by averaging three independently calculated normalized expression values for each sample.

### 4.5. Anthocyanin Extraction from Small Samples

Plant tissues or cell cultures were harvested and freeze-dried at −80 °C. Before extraction, the samples were ground in a chilled mortar and pestle to enhance the extraction process. The extraction ratio was 1 g of DW samples/5 mL of buffer to normalize the results on the dry weight of the samples. The extraction process was performed in an aqueous buffer with citric acid 4 g/L and ascorbic acid 1 g/L, and the mixtures were centrifuged using a refrigerated centrifuge at 5000× *g* rpm for 10 min. The prepared extracts were then filtered through a 0.20 µm membrane filter before HPLC analysis.

### 4.6. Anthocyanin Extraction, Purification, and Lyophilization from a Bioreactor

After 14 days of culture in a bioreactor, the cells were harvested by physical separation from the medium and lysed in a citric acid 4 g/L and ascorbic acid 1 g/L buffer overnight at 4 °C. The following day, they were filtered by a 0.2 µm tangential flow filtration. The permeate was then passed in a column of 3.5 cm in diameter and 50 cm in length through 1 L of hydrophobic resin (HPD 450) packed by gravity force. Thanks to hydrophobic interactions, anthocyanins bind the resin particles, so the metabolite was then eluted with absolute ethanol and then removed with a rotavapor. Anthocyanins were frozen at −80 °C overnight and then lyophilized.

### 4.7. HPLC and MS Analysis

The lyophilized powder was dissolved in water with 0.2% acetic acid to obtain a final 0.4 mg/mL solution. The analysis was performed using a High-Performance Liquid Chromatography system (1200 series HPLC system Chip Cube nano-ESI interface) coupled with a 6520 Quadrupole Time-of-Flight (Q-TOF) Mass Spectrometer (Agilent Technologies, Santa Clara, CA, USA). The Q-TOF mass spectrometer was equipped with a nano-electrospray ionization source (nano-ESI). For the chromatography, a Large Capacity Chip 160 nL enrichment column and a column of 0.5 × 150 mm Zorbax 300SB-C18 5 µm were used. The Mobile phase of the nano pump was A = H_2_O:MeOH 97:3 + 0.1%FA; B = ACN:MeOH:H_2_O 90:5:5 + 6mM NH4FA + 0.1%FA. Its gradient parameters were 0 min (10% B), 0–13 min (70% B), 13–14.5min (84% B), 14.5–20 min (84% B), 20–23 min (10% B), and 23–32 min (10% B), with a flow rate of 0.4 µL/min. The mobile phase of the cap pump was A = H_2_O:MeOH 97:3 + 0.1%FA; B = acetonitrile. Its gradient parameter was isocratic 10% B, with a flow rate of 2 µL/min. The injection volume was 0.5 µL.

For the Mass Spectrometry parameters, the Q-TOF operated in MS mode at 2 GHz extended dynamic range, with three reference ions. The capillary voltage was set to 1700 V (positive polarity) with nitrogen as the desolvating gas at 330 °C and 4.8 L/min; fragmentor, skimmer1, and octupole were set at 130, 65, and 750 V, respectively. Mass spectra were acquired in a data-dependent mode: the MS/MS spectra of the seven most intense ions were acquired for each MS scan in the 140–1700 Da range, with a narrow isolation width of 1.3 amu. The scan speed was set to 2 MS spectra/sec and 4 MS/MS spectra/sec.

The characterization process of cyanidin and anthocyanidins was based on their mass spectral data. The first approach was to search compounds by the aglycone fragment of cyanidin, delphinidin, petunidin, pelargonidin, peonidin, and malvidin as [M]^+^ at *m*/*z* 287.055, 303.0499, 317.0656, 271.0601, 301.0706, and 331.0812, respectively. Each chromatographic peak was manually explored to verify the *m*/*z* of precursor ions and fragmentation spectra at different collision energies.

### 4.8. Cell Viability Assay

THP1 cells (human monocytes; TIB-202™ ATCC, distributed by LGC Standards, Milan, Italy) were cultured in RPMI-1640 medium containing 2-mercaptoethanol 0.05 mM, heat-inactivated fetal bovine serum (FBS) 10% *v*/*v*, and glutamine 2 mM (all provided by Thermo Fisher, Milan, Italy). For the described experiments, THP-1 cells were differentiated into cells with macrophage-like phenotype by incubating in 5 ng/mL phorbol 12-myristate 13-acetate (PMA; Merck, Milan, Italy) for 72 h [36]. BJ cells (human fibroblast cell line; CRL-2522™ ATCC) were cultured in Eagle’s Minimum Essential Medium containing penicillin/streptomycin 1% *v*/*v*, FBS 10% *v*/*v*, and glutamine 2 mM (all provided by Thermo Fisher). The cells were cultured at 37 °C with 5% CO_2_ humidified air.

The lyophile (ANT-CA) was suspended in dimethyl sulfoxide (DMSO) at 200 mg/mL and kept at room temperature in the dark. Working solutions of ANT-CA were prepared in cell culture media and stored at 4 °C in the dark.

Cell viability was measured using the 3-(4,5-di methyl thiazol-2-yl)-2,5-diphenyltetrazolium bromide (MTT, Merck) assay according to the method proposed by Denizot et al. [37], with minor modifications. Briefly, cells (1 × 10^5^ cells/mL) were cultured in 96-well plates (Sarstedt Srl, Milan, Italy) and incubated for 24 h. Culture media were renewed and cells were incubated with ANT-CA 0–200 μg/mL for 24 h. Cells were then incubated for 4 h at 37 °C with 5 mg/mL of MTT solution. Formazan crystals were solubilized in 100 µL of sodium dodecyl sulfate (SDS) 10% *vol*/*vol*–HCl 0.01 N for 16 h. Optical density (O.D.) was recorded at 570 nm using a microplate reader (Varioskan Lux Reader, Thermo Fisher Scientific). The cell viability was calculated as % viability = (sample absorbance/control absorbance) × 100.

### 4.9. Intracellular Antioxidant Activity

THP-1 and BJ cells were seeded in 24-well tissue culture plates (Sarstedt Srl) at 3 × 10^5^ cells/mL. Twenty-four hours later, cells were loaded for 30 min at 37 °C with 10 mM 2′,7′-dichlorodihydrofluorescein diacetate (H_2_DCFDA; Molecular Probes, Thermo Fisher) in warm PBS [38]. At the end of the incubation, cultures were washed twice and incubated with ANT-CA for 2 and 4 h at 37 °C. The culture media was renewed and the cells were added with H_2_O_2_ (100 μM) and incubated for an additional 30 min. To quantify dichlorofluorescein (DCF) fluorescence intensity, cells were lysed and fluorescence in the supernatant was measured using a microplate reader (Varioskan Lux Reader) at ex 485 nm, em 530 nm [39].

### 4.10. Anti-Inflammatory Activity

THP-1 cells (3 × 10^5^ cells/mL) were incubated for 4 h with 100 μg/mL of ANT-CA and then challenged for an additional 20 h with lipopolysaccharide (LPS from *Salmonella enterica* serotype *enteritidis*, 100 ng/mL; Merck). Interleukin (IL)-1β and IL-6 were measured in the conditioned medium using commercially available ELISA kits (e-Bioscience, Prodotti Gianni, Milan, Italy). Optical densities were measured at 450 nm using a microplate reader (Varioskan Lux Reader). The sensitivity of the assays ranged between 1 and 15 pg/mL.

### 4.11. Wound Healing Assay

BJ cells (3 × 10^5^ cells/mL) were seeded onto glass coverslips and cultured at 37 °C for 24 h. Cells were treated with ANT-CA 100 μg/mL and the cell monolayers were then wounded with a plastic tip. Seventy-two hours later, the cells were rinsed twice in PBS, fixed in paraformaldehyde (PFA) 4% *w/vol* for 10 min, and routinely stained with hematoxylin and eosin. The slides were examined using a light transmission microscope connected to a camera to capture the images (DMLB Leica, Wetzlar, Germany).

### 4.12. Antibacterial Activity

Methicillin-resistant *Staphylococcus aureus* (MRSA; ATCC 33592) and *Pseudomonas aeruginosa* (*P. aeruginosa*; ATCC BAA-2108) were purchased from ATCC (LGC Standards; Milan, Italy). Bacteria were cultured on Mueller–Hinton agar (MHA) or broth (Fisher Scientific, Milan, Italy). The antibacterial activity of ANT-CA was assessed using the agar well diffusion method. Overnight cultures of MRSA or *P. aeruginosa* (10^4^ CFU/mL) were spread on MHA plates using sterile bacteriological glass rods. Then, wells were made in the agar using a sterile cork borer, and 100 μL ANT-CA (100 µg/mL) was added per well. Plates were incubated at 37 °C for 24 h. The presence of clear zones indicated antibacterial activity. The clear zones were measured using a ruler and the data were graphed. Positive controls (wells containing 0.1 μg/mL daptomycin or ciprofloxacin) were prepared in each experiment.

### 4.13. Statistical Analysis

The data are reported as mean values ± the standard error of the mean (SEM). Statistical analysis was performed using GraphPad Prism 9.5.1 Software (GraphPad, San Diego, CA, USA). Statistical differences were assessed by *t*-test or one-way ANOVA followed by the Bonferroni multicomparison post hoc test. Statistical significance was considered for *p*-values of 0.05 or less.

## Figures and Tables

**Figure 1 ijms-24-13711-f001:**
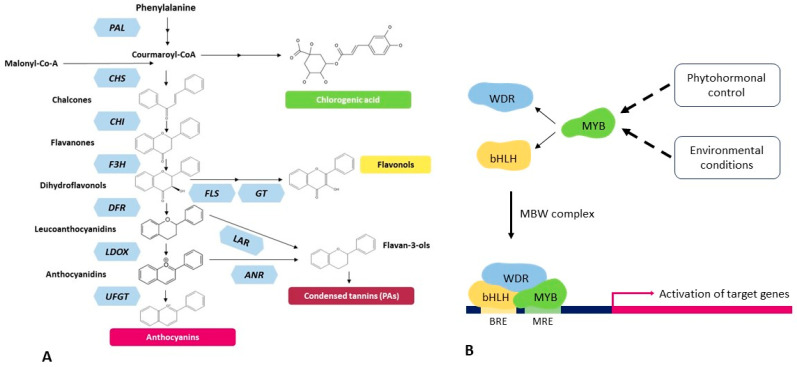
Schematic diagram of the flavonoid biosynthetic pathway (**A**) and its regulation by the MBW complex (**B**) in plants. (**A**) The enzymes shown in light blue hexagons, while reactions products are drawn in black: PAL phenylalanine ammonia-lyase, CHS chalcone synthase, CHI chalcone isomerase, F3H flavanone-3β-hydroxylase, FLS flavonol synthase, GT unidentified enzyme encoding a glycosyl transferase for flavonol glycone synthase, DFR dihydroflavonol 4-reductase, LAR leucoanthocyanidin reductase, LDOX leucoanthocyanidin dioxygenase, ANR anthocyanidin reductase, and UFGT UDP-glucose:flavonoid-3-O-glucosyltransferase. (**B**) The transcriptional regulators bHLH and MYB bind the DNA at specific responsive elements (BRE and MRE, respectively), while the WDR protein stabilizes the MBW complex.

**Figure 2 ijms-24-13711-f002:**
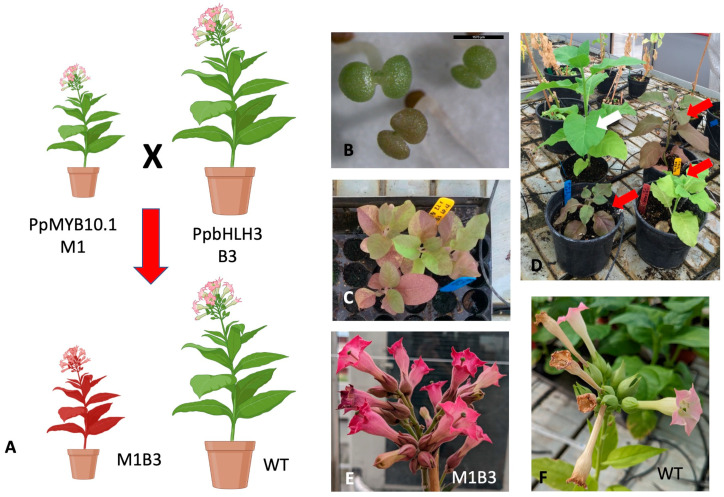
Development of tobacco M1B3 plant clones. Plants carrying the PpMYB10.1 (M1) peach cDNA were crossed with clones carrying the PpbHLH3 (B3) cDNA to obtain siblings expressing both TFs with a LhG4/pOp expression system (**A**) (created with BioRender.com). M1B3 plantlets were selected at the cotyledonary stage (**B**) for their color, further selected on soil (**C**), and grown under standard greenhouse conditions with controls (WT, white arrow), showing different amounts of pigment accumulation (M1B3 plants, red arrows) (**D**). Anthocyanin accumulation was particularly strong in the flowers of M1B3 plants (**E**) compared to WT (**F**).

**Figure 3 ijms-24-13711-f003:**
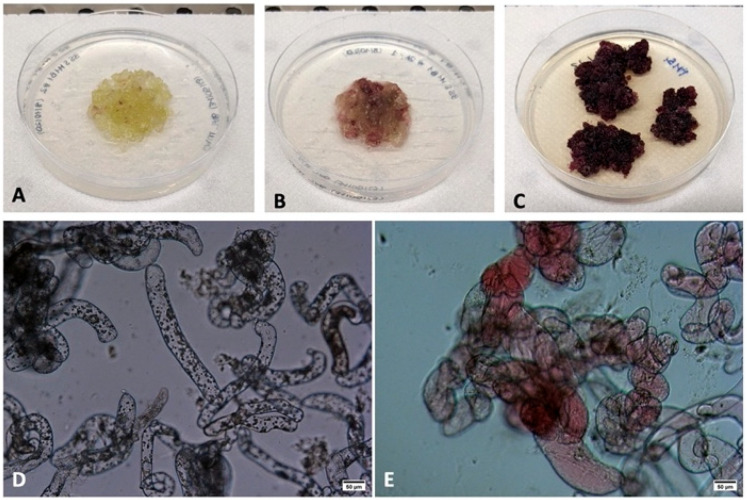
Development of tobacco M1B3 plant cell lines. Cell lines were developed from the leaves of individuals showing strong pigmentation (**A**). A further selection of the best genotypes was performed at the cell culture level on solid media (**B**,**C**). From the best calluses, liquid cultures were developed. At the end of the culture period, cells of both M1B3 (**E**) and control (**D**) genotypes showed a rod-like structure, but anthocyanins were detectable only in M1B3 cells (bar in (**D**,**E**): 50 µm).

**Figure 4 ijms-24-13711-f004:**
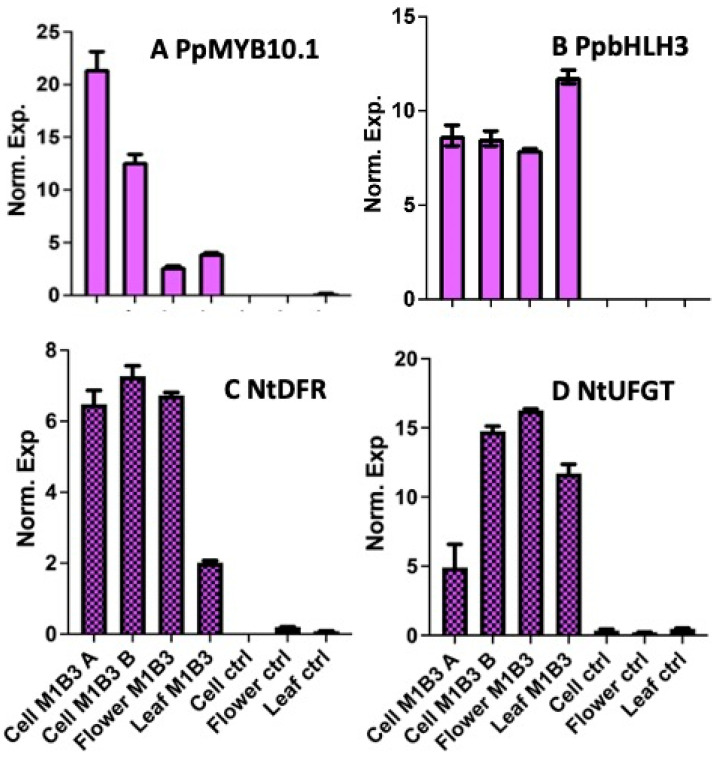
Molecular characterization of tobacco M1B3 plant cell lines. The expression of the PpMYB10.1 (**A**) and PpbHLH3 (**B**) peach transgenes and of two possible tobacco target genes (NtDFR -C- and NtUFGT -D-) was measured by qRT-PC, in the leaves and flowers of the clone used to develop the cell lines (“Leaf M1B3” and “Flower M1B3”, respectively), in two batches of cells at the end of the culture period (“Cell M1B3 A” and “Cell M1B3 B”, respectively) and in cells grown in culture “Cell ctrl”. Also the organs of a greenhouse-grown plant of a control (ctrl) genotype have been tested (“Flower ctrl” and “Leaf ctrl” for flowers and leaves, respectively). Column/sample names in (**A**) and (**B**) as in (**C**) and (**D**), respectively. The values are the means of the normalized expression and the bars are the standard deviations from the means of three replicates.

**Figure 5 ijms-24-13711-f005:**
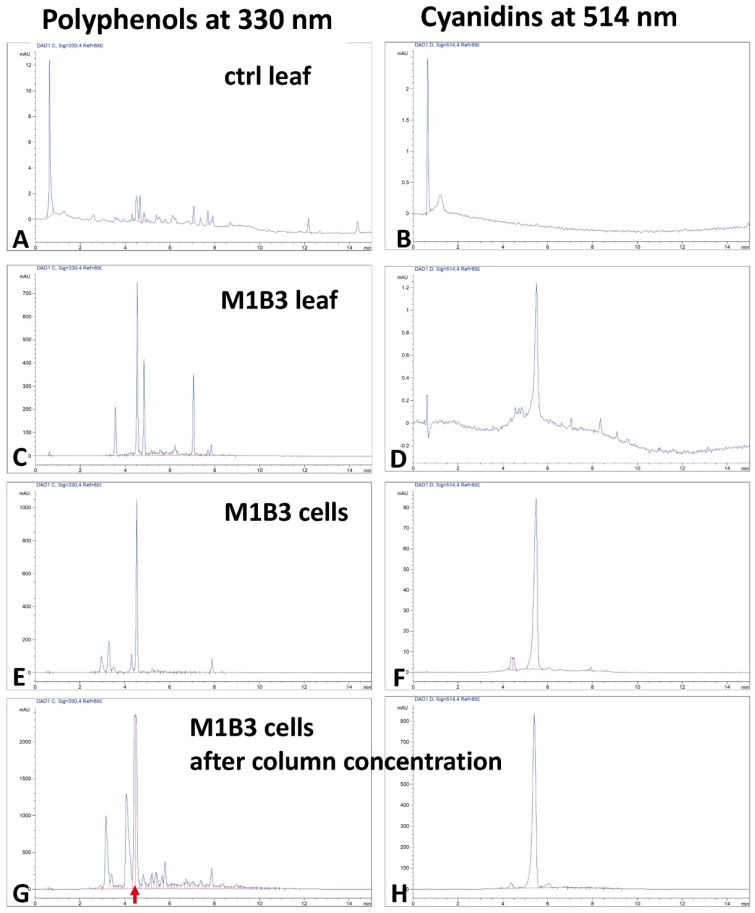
HPLC characterization of the polyphenol extracts. Panels on the left are for total polyphenols (detector set at 330 nm), while those on the right are for anthocyanins (detector set at 514 nm). Samples are normalized by the dry weight of the starting material and are leaf extracts from control (ctrl leaf, (**A**,**B**)) and transgenic (M1B3 leaf, (**C**,**D**)) plants; cell extracts from cultured cells at the end of the growing period (M1B3 cells, (**E**,**F**)); and lyophile obtained from the industrial-scale purification column (M1B3 cells after column concentration, (**G**,**H**)). The red arrow highlights the elution time for chlorogenic acid highlighting the corresponding peaks in panels (**C**,**E**,**G**).

**Figure 6 ijms-24-13711-f006:**
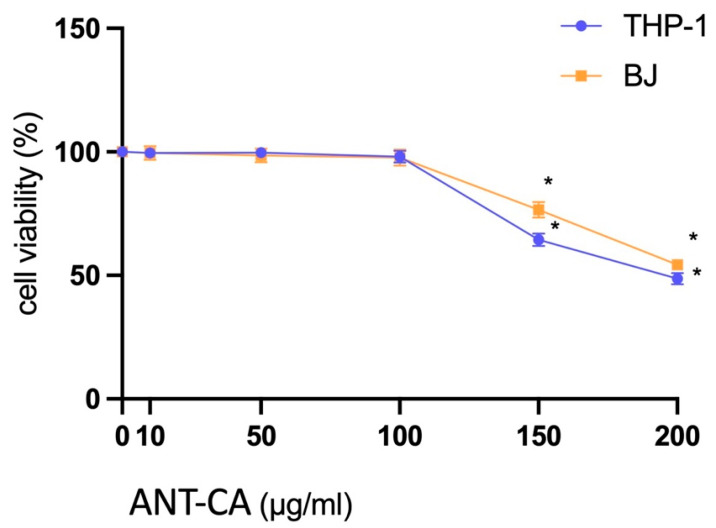
Viability of cells treated with ANT-CA. THP-1 (blue line) and BJ (orange line) cell lines were treated for 24 h with ANT-CA at concentrations ranging from 0 to 200 μg/mL. Cell viability was determined by an MTT assay and calculated as the percentage of viability over non-treated cells. Data are reported as mean ± SEM of 3 independent experiments, each performed in triplicate. * denotes *p* < 0.05 vs. non-treated cells.

**Figure 7 ijms-24-13711-f007:**
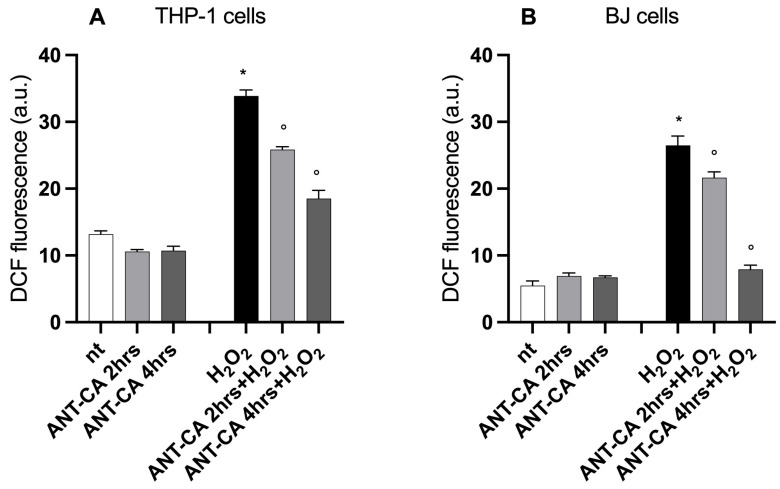
Antioxidant activity of ANT-CA. THP-1 (**A**) and BJ (**B**) cells were treated for 2 or 4 h with 100 μg/mL ANT-CA and then challenged with H_2_O_2_ at 100 μM for 30 min. At the end of incubation, cells were lysed and intracellular fluorescence was quantified using a microplate reader at ex 485 nm/em 530 nm. Data are reported as mean ± SEM of 3 independent experiments, each performed in duplicate. * denotes *p* < 0.05 vs. non-treated (nt) cells; ° denotes *p* < 0.05 vs. H_2_O_2_ treated cells.

**Figure 8 ijms-24-13711-f008:**
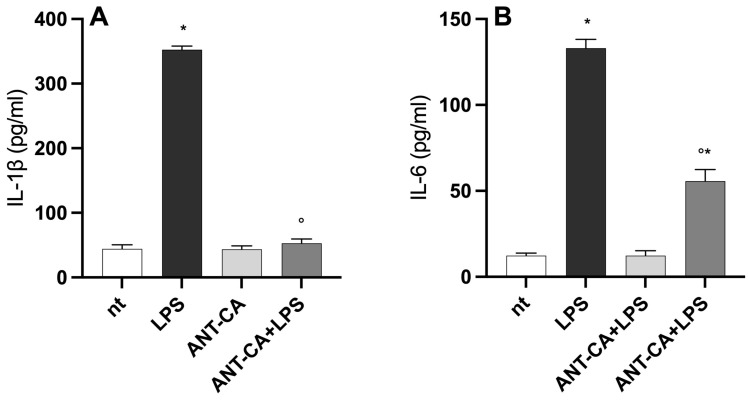
Anti-inflammatory activities of ANT-CA. THP-1 cells were treated for 4 h with 100 μg/mL ANT-CA and then incubated for 20 h with lipopolysaccharide (LPS; 100 ng/mL). Interleukin (IL)-1β (**A**) and IL6 (**B**) were measured in the conditioned media using commercially available ELISA kits. Data are reported as mean ± SEM of 3 independent experiments, each performed in duplicate. * denotes *p* < 0.02 vs. non-treated (nt) cells; ° denotes *p* < 0.05 vs. LPS treated cells.

**Figure 9 ijms-24-13711-f009:**
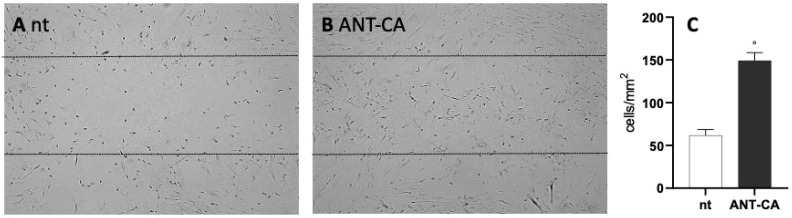
ANT-CA supports wound healing. (**A**) BJ cells were grown at confluency onto glass coverslips. (**B**) ANT-CA 100 μg/mL was then added to the cells and the cell monolayers were wounded. After 72 h, the cells were fixed in PFA, stained with hematoxylin and eosin, and then visualized using a light transmission microscope (10× objective) equipped with a camera. The representative results are of three independent experiments. (**C**) Following 72 h of incubation, the number of cells crossing the lines was evaluated and normalized by the analyzed area. Data are reported as mean ± SEM of 3 independent experiments, each performed in duplicate. ° denotes *p* < 0.05 vs. non-treated (nt) cells.

**Figure 10 ijms-24-13711-f010:**
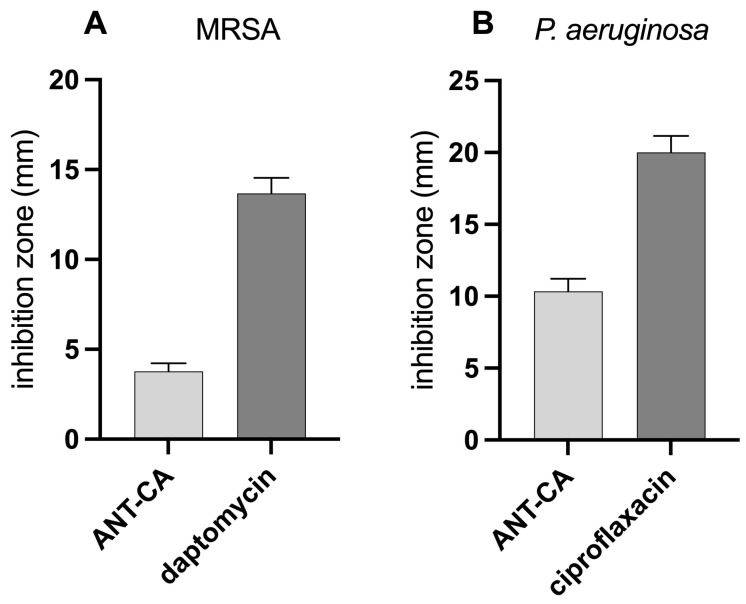
Antibacterial activity of ANT-CA. (**A**) Methicillin-resistant *Staphylococcus aureus* (MRSA) and (**B**) *Pseudomonas aeruginosa* (*P. aeruginosa*) were spread (10^4^ CFU/mL) on agar plates. ANT-CA (100 µg/mL) and antibiotics (0.1 μg/mL daptomycin for MRSA; 0.1 μg/mL ciprofloxacin for *P. aeruginosa*) were dropped into wells made in the agar. The plates were incubated at 37 °C for 24 h. Clear zones indicative of bacterial growth inhibition were measured. Data are reported as mean ± SEM of 6 independent experiments, each performed in triplicate.

**Table 1 ijms-24-13711-t001:** Anthocyanin compounds identified by nanoHPLC-MS/MS in the analyzed ANT-CA lyophile. Rt: retention time.

MS Ion [M]^+^ (*m*/*z*)	MS/MS Ions [M]^+^ (*m*/*z*)	Theoretical MS [M]^+^ (*m*/*z*)	ppm	Rt (min)	% Anthocyanins (Peak Area/Total Peak Area)	Name	Formula
595.1650	287.0569, 449.0889	595.1657	1.2	4.02	72.5	cyanidin 3-rutinoside	C_27_H_31_O_15_^+^
627.1507	303.0434, 465.0923	627.1555	7.6	5.77	10.3	delphinidin 3,5-diglucoside	C_27_H_31_O_17_^+^
611.1566	303.0434, 465.0923	611.1606	6.5	6.6	10.0	delphinidin 3-rutinoside	C_27_H_31_O_16_^+^
789.2044	627.1507, 465.0901	789.2084	5.1	2.12	0.7	delphinidin 3,5,3’-triglucoside	C_33_H_41_O_22_^+^
641.1721	317.0614, 479.1162	641.1712	1.4	5.21	3.8	petunidin 3,5 diglucoside	C_28_H_33_O_17_^+^
317.0687	302.0363, 285.0330	317.0655	10	7.43	0.6	petunidin	C_16_H_13_O_7_^+^

**Table 2 ijms-24-13711-t002:** List of primers used for gene expression analyses by qRT-PCR.

Gene Name	Primer	Sequence (5′ → 3′)	Accession no. (Phytozome/GeneBank)
*MYB10.1*	For	CAGGAAGGACAGCGAATGATG	ppa026640m
	Rev	TCGGGGTTGAGGTCTTATTACG	
*bHLH3*	For	TCTTGTTCAGCGTTCCGTTCCT	ppa002884m
	Rev	TTGGCGCTGAGCTCATCTTGTG	
*NtDFR*	For	CAGAGAAGGCCGCAATGGAAGC	AB289448
	Rev	GGTGGGAATGTAGGCGTGAGGAAT	
*NtUFGT*	For	CAATGAGTGCATTGGATGCC	FG627024
	Rev	CCAGCTCCATTAGGTCCTTG	
*NtUBI*	For	AGGGAAGCAACTTGAAGATGGA	XM_016614723
	Rev	CCCCTCAAACGCAACACC	

## Data Availability

Not applicable.

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
