# Peer review of "Optimization of Anthocyanin Production in Tobacco Cells"

_ijms, 2023, doi:10.3390/ijms241813711_

Round 1

Reviewer 1 Report

The present manuscript describes development of transgenic anthocyanin overproducing cell line of Nicotiana tabacum through co-expression of MYB and bHLH transcription factors. The authors demonstrate large scale production of anthocyanin and other phenolics in the cell line. Finally authors show the isolation of lyophile, called ANT-CA, which displays health beneficial bioactivities. The study is interesting and shows the feasibility of cell cultures for large scale production of anthocyanin and related compounds.  However, there are certain comments, which authors need to address

1. The expression analysis of only three structural genes of flavonoid biosynthesis has been carried out. What is about the expression of genes like PAL, CHS, which play an important role in the substrate flux channeling towards anthocyanin biosynthesis. Expression analysis of these genes should be carried out.

2.  One earlier study by Zhou et al. 2008 (Planta. 2008 Dec; 229(1):37-51) also demonstrated anthocyanin production in transgenic tobacco cultures. The authors should discuss considering the results of this study as well.

3. What is the driving construct? It is important to provide a brief detail for better understanding.

4. Figure 2:  The schematic representation shows the picture of N. benthamiana. But the experimental system is Nicotiana tabacum.  The schematic representation should be modified accordingly. Otherwise it will confuse the audience.

5. There are grammatical mistakes and typos, that should be checked an edited thoroughly.

6. Line 152: “Compared to the leaves” which leaf? Pigmented or WT?

7. Line 158-159: There is something missing in the sentence.

8. Line 256: “Looked green” ?

9. Section 2.3 2.3. “Cell viability assay with the ANT-CA”. There should be some context before detailing the results in the section.

In general, manuscript is well-written. However, there are several grammatical mistakes and typos, which should be corrected. 

Reviewer 2 Report

The presented manuscript is devoted to the optimization of the production of anthocyanins. the authors managed to obtain tobacco cells with a high content of anthocyanins. This is a big success. Next, the authors tried to study the composition of ANT-CA, the authors determined its main components. However, studying the properties of multicomponent ANT-CA is not the best option, so the authors used a high concentration of ANT-CA. For example,

when determining the antibacterial activity, the concentration of ANT-CA -100 mkg / ml was used, and antibiotics - 0.1 mkg / ml, this is a low antibacterial activity, not a moderate one.

82 - Indicate the purpose of the present study, not the results obtained

Fig. 4 - indicate a statistically significant coefficient

Table 1 - specify sample (ANT-CA -?).

Fig. 9 - hard to see, no difference between nt and ANT-CA

212 - decrypt LPS

286 - links to the activity of the drug ANT-CA. Has it been previously characterized?

377 - space

In the Discussion section, put more emphasis on discussing the results

Round 2

Reviewer 1 Report

The authors have adequately revised the manuscript, and satisfied most of my concerns.